# Long Non-Coding RNAs as Novel Biomarkers in the Clinical Management of Papillary Renal Cell Carcinoma Patients: A Promise or a Pledge?

**DOI:** 10.3390/cells11101658

**Published:** 2022-05-17

**Authors:** Francesco Trevisani, Matteo Floris, Riccardo Vago, Roberto Minnei, Alessandra Cinque

**Affiliations:** 1Urological Research Institute, San Raffaele Scientific Institute, 20132 Milano, Italy; vago.riccardo@hsr.it; 2Unit of Urology, San Raffaele Scientific Institute, 20132 Milano, Italy; 3Biorek s.r.l., San Raffaele Scientific Institute, 20132 Milano, Italy; alessandra.cinque@biorek.eu; 4Nephrology, Dialysis, and Transplantation Division, G. Brotzu Hospital, University of Cagliari, 09134 Cagliari, Italy; matteo.floris@aob.it (M.F.); rob.minnei@gmail.com (R.M.)

**Keywords:** papillary renal carcinoma, diagnosis, prognosis, molecular biomarkers, non-coding RNA, long non-coding RNA, ferroptosis

## Abstract

Papillary renal cell carcinoma (pRCC) represents the second most common subtype of renal cell carcinoma, following clear cell carcinoma and accounting for 10–15% of cases. For around 20 years, pRCCs have been classified according to their mere histopathologic appearance, unsupported by genetic and molecular evidence, with an unmet need for clinically relevant classification. Moreover, patients with non-clear cell renal cell carcinomas have been seldom included in large clinical trials; therefore, the therapeutic landscape is less defined than in the clear cell subtype. However, in the last decades, the evolving comprehension of pRCC molecular features has led to a growing use of target therapy and to better oncological outcomes. Nonetheless, a reliable molecular biomarker able to detect the aggressiveness of pRCC is not yet available in clinical practice. As a result, the pRCC correct prognosis remains cumbersome, and new biomarkers able to stratify patients upon risk of recurrence are strongly needed. Non-coding RNAs (ncRNAs) are functional elements which play critical roles in gene expression, at the epigenetic, transcriptional, and post-transcriptional levels. In the last decade, ncRNAs have gained importance as possible biomarkers for several types of diseases, especially in the cancer universe. In this review, we analyzed the role of long non-coding RNAs (lncRNAs) in the prognosis of pRCC, with a particular focus on their networking. In fact, in the competing endogenous RNA hypothesis, lncRNAs can bind miRNAs, resulting in the modulation of the mRNA levels targeted by the sponged miRNA, leading to additional regulation of the target gene expression and increasing complexity in the biological processes.

## 1. Introduction

Papillary renal cell carcinoma (pRCC) represents the second most common histology of renal cancer, following clear cell renal carcinoma and accounting for 10–20% of all renal cell cancers. In the last years, there has been a remarkable progress in the explanation of the molecular basis of this type of neoplasm. However, a reliable molecular biomarker able to detect the presence and the grade of malignancy of pRCC is not yet available in day-to-day clinical practice. Long-non coding RNA (lncRNAs) represents a new promising family of non-coding RNAs which plays several important biological roles in the regulation of miRNAs and their mRNAs targets [1]. Our review summarizes the most promising lncRNA signatures involved in the diagnosis and prognosis of patients affected by pRCC, highlighting their molecular networking and suggesting their possible implication in the clinical management as a predictive tool able to stratify patients upon cancer aggressiveness.

## 2. Epidemiology and Risk Factors

### 2.1. Epidemiology

Renal cancer (RC) is the second most common and deadly neoplasm of the urinary tract, following bladder cancer. According to the latest GLOBOCAN report, in 2020, there were 421,288 new cases of RC and 179,368 deaths worldwide, being the 14th most incident (2.2% of new cases of all-sites cancers) and 15th most deadly cancer type (1.8% of new deaths from all-sites cancers) [2].

RC incidence increases with the Human Development Index, and half of cases are diagnosed in Europe and North America, [2].

RC incidence has increased over time [3]. Over a 27-year period, from 1990 to 2007, ASRs increased from 4.72/100,000 to 4.94/100,000 worldwide; however, the incidence is expected to decrease in developed countries to an ASR of 4.46/100,000 within 2030 [4]. In the United States, from 1973 to 1998, the incidence of localized, regional, and metastatic renal cell carcinoma (RCC) increased (3.7/100,000, 1.9/100,000 and 0.68/100,000 annual percentage change, respectively), supporting a contributing role both of the increased use of abdominal imaging and non-imaging-related risk factors (RFs) [5,6].

RC displays a male predominance, with a M:F ratio ranging from 1,69:1 to 3:1 [2,7].

The mean age at presentation is between 60 and 65 years old [8,9] but certain histotypes or syndromic forms of RC usually have an earlier onset, e.g., papillary RCC (pRCC) in hereditary papillary RCC has a mean age at onset of 41 years [10].

US data showed that RC incidence is similar among ethnicities, except for Asian and Pacific Islanders, who displayed lower incidence rates [11].

Black patients have a lower survival rate than White patients with a similar mortality, since the former are more frequently diagnosed with less aggressive histotypes of RCCs (pRCC or chromophobe renal cell carcinoma (chRCC) than with ccRCC) [3].

### 2.2. Risk Factors

Several RFs have been identified for the various types of RCs. Renal cell carcinoma is associated with genetic factors, as is the case in hereditary cases of pRCCs (hereditary papillary renal cancer, hereditary leiomyomatosis and RCC, rarely Birt–Hogg–Dubè syndrome) [12,13,14] and acquired factors. The latter include common cardiovascular RFs such as smoking, hypertension and obesity [15,16]; renal disease, chronic kidney disease (CKD) and end stage renal disease (ESRD) (including dialysis and transplanted patients) [15,17,18,19] and cytotoxic chemotherapy [20], kidney stones [21,22]; occupational agents, such as trichloroethylene (IIa or probable human carcinogen, according to the IARC) [23,24]; prolonged analgesic assumption [25,26]; and chronic hepatitis C virus infection [15,27].

## 3. Classification of Renal Cancers, Macroscopic and Microscopic Anatomy, Grading

### 3.1. Classification of RC and RCC

Renal cancers are usually classified according to the most recent World Health Organization (WHO) classification of tumors. In 2016, it identified diverse categories of renal cancer based on histogenesis and age at onset criteria; pRCC is classified as a “renal cell tumor” [28]. Renal cell tumors are further classified according to several criteria (according to morphology, histogenesis, location, disease association, and genetic abnormality) [28]. The WHO classification of kidney tumors and the subclassification of renal cell tumors can be found in Table 1.

### 3.2. Classification of pRCC

pRCC is the second most common RCC, accounting for 10–15% of cases [7,29]. The precise histogenesis has not yet been determined; however, it is believed to originate from the proximal tubule epithelial cells, as is the case with ccRCC [30].

Although sporadic descriptions of papillary renal cancer date back to the 1950s [31] and 1960s [32], pRCC was firstly and comprehensively described by Mancilla-Jimenes in 1976 [33,34]. The traditional distinction in type 1 and type 2 pRCC was formally described by Delahunt and Eble in 1997 [35]. pRCCs were officially recognized as a distinct entity in the Heidelberg Classification in 1997 [36], leading to their inclusion in the WHO classification in 2004 [37].

According to the 2013 ISUP Vancouver classification of renal neoplasia, the subdivision into the two pRCC types is of value [38]. In a large retrospective study published in 2016, 161 pRCCs were molecularly characterized, and the two subtypes appeared as two distinct entities from a clinical and biological standpoint. Moreover, type 2 pRCC was further classified into at least three subtypes according to the molecular features and their association with patient survival [39].

Traditional morphologic subdivision into type 1 and type 2 is not always feasible, and the morphologic criterion remains controversial because of the lack of molecular and biochemical evidence [38]. In the last 20–30 years, our knowledge of the molecular abnormalities featured in pRCCs has grown, leading to a deeper understanding of the morphological and biological spectrum of renal tumors with papillary growth pattern [40]. As a matter of fact, the 2021 GUPS update on renal neoplasia stated that pRCC subtyping is no longer recommended because of the subjectivity in applying the histologic criteria, overlapping features among subtypes, and lack of clinical benefit. Moreover, histologic architecture and grading (according to WHO/ISUP grading system) are better prognostic factors than pRCC subtyping. Novel pRCC patterns were identified, namely, biphasic, solid, Warthin-like, and papillary neoplasms with reverse polarity [41].

With regard to pRCC, some renal entities are bound to pRCC due to their similar microscopic appearance and/or molecular features, for instance, clear cell tubulopapillary RCC, tubulocystic-RCC, and mucinous tubular and spindle carcinoma [41,42]. After the publication of the most recent WHO classification in 2016, provisional or emerging tumor entities with papillary growth were described, including papillary renal neoplasm with reversed polarity, biphasic hyalinizing psammomatous RCC, and biphasic squamoid/alveolar RCC [40].

### 3.3. Molecular and Genetic Features

The two subtypes of pRCC are different molecular entities, and pRCC can be considered a heterogeneous spectrum of neoplasms [43]. The cancer genome atlas (TCGA) study of pRCC identified four sub-groups of tumors according to their molecular alterations [39]. Type 1 pRCCs are typically associated with activating mutations of the MET gene (located at chromosome 7q31.1-34) in a subset of sporadic cases [44,45], usually leading to the activation of the tyrosine kinase domain of the MET protein, a membrane receptor for hepatocyte growth factor (HGF) [39,46]. Of the two subtypes, type 1 pRCCs are usually associated with at least one chromosomal abnormality, with chromosome 7 trisomy [47] and chromosome 17 trisomy being observed in 70–80% of cases, and less frequently, a gain of other chromosomes (2, 3, 12, 16, and 20), a genetic pattern also found in hereditary papillary renal cancer, characterized by multifocal, bilateral type 1 pRCC [39]. Another common alteration is the loss of Y chromosome in males [48].

On the contrary, type 2 usually features different and heterogeneous alterations (CDKN2A silencing, SETD2 mutations, TFE3 fusions, and increased expression of the NRF2-antioxidant response element pathway, mutations in chromatin modifying genes such as SETD2, BAP1 and PBRM1) [48]. Furthermore, only type 2 tumors exhibit loss of chromosome 1p, loss of 3p, and gain of 5q [49]. In a TCGA study, type 2 pRCCs were classified into three subtypes from a molecular standpoint. Furthermore, a CpG island methylator phenotype (CIMP-RCC) was observed in a subgroup (5.6%) of type 2 tumors often harboring a fumarate hydratase gene mutation, characterized by the worst survival rate in the study population [39]. Fumarate hydratase mutations can be found in a hereditary form of type 2 pRCC, in hereditary leiomyomatosis and in renal cell carcinoma. FH gene is considered a housekeeping gene; however, the oncogenic mechanisms are still unclear [48]. Pal et al. investigated pRCC molecular features in a population that included mostly patients with stage IV disease, contrarily to the TCGA study. Compared to the findings of the latter study, they found a similar spectrum of mutations for the two pRCCs subtypes, but 7% of type 2 pRCC notably displayed a MET mutation that, with CDK2N2A/B mutations, might represent a potential therapeutic target in metastatic pRCCs [50].

### 3.4. Macroscopic and Microscopic Anatomy

pRCC presents with a mean maximum mass of 5.3–7.6 cm, although the recent trend characterized by incidental early-stage diagnoses of small renal masses has led to downsizing of the mean dimensions at diagnosis [51,52,53]. Hemorrhage, necrosis and cystic changes are common features [54]; moreover, pRCC is the renal cell carcinoma that is most commonly associated with a fibrous pseudocapsule [52].

Neoplastic cells characteristically line a fibrovascular structure in the papillary pattern, but a tubulopapillary growth pattern with papillae and coexisting compact tubules can also be observed [35,55]. Other patterns, such as solid and solid–glomeruloid patterns, enrich the wide spectrum of tumor architectures [56]. Novel pRCC patterns were recently described, namely biphasic, solid, Warthin-like, and papillary neoplasms with reverse polarity [41].

In light microscopy examination, type 1 pRCCs are characterized by a basophilic cuboidal or columnar cell, with round regular nuclei, with the presence of psammoma bodies and foci of foam cells. On the contrary, type 2 pRCCs exhibit larger eosinophilic cells, with irregular nuclei with prominent nucleoli, typical of a higher grading [28,35].

On immunohistochemistry, the two subtypes have different patterns; type 1 pRCC is immunoreactive for CK7, vimentin and MUC1, AMACR and CD10, while type 2 tumors stain positive with CK20, E-cadherin, AMACR and CD10, but can stain negative for CK7, unlike type 1 tumors [52,57,58].

### 3.5. Grading

The Fuhrman grading system assesses nuclear size, nuclear pleomorphism and nucleolar prominence, and it was traditionally used to grade renal cell carcinomas. However, due to its suboptimal reproducibility [59,60,61,62], and the unclear prognostic significance in pRCCs and other non-ccRCCs [55,63], the 2019 European Society for Medical Oncology (ESMO) and the 2021 European Association of Urology (EAU) guidelines on RCC recommend the use of the 2012 ISUP grading system over the Fuhrman grading system [64,65]. The former is a four-tier RCC grading system based on the highest grade of abnormality exhibited (nucleoli prominence, nuclear pleomorphism, presence of tumor giant cells or sarcomatoid and/or rhabdoid differentiation) [64], whose prognostic value has been proven in pRCCs patients [62,66,67,68] (Table 2). The 2012 WHO/ISUP grading system is used to systematically classify the grading of pRCCs because of the association between grades and the biological behavior of pRCCs [69].

## 4. Presentation at Diagnosis, Clinical Course, Prognosis

### 4.1. Clinical Manifestations

pRCCs are more commonly diagnosed between the 6th and 8th decades, compared to RCCs, and male and black individuals have a greater pRCC incidence [3]. Disease onset before 46 years of age may indicate a hereditary syndrome and should prompt a genetic evaluation, according to the American Society of Clinical Oncology [10]. An inherited form should be suspected with multifocal, bilateral, and early onset hypovascular renal masses [70].

Incidental RCC diagnoses currently account for 40–50% of cases [53,71,72,73].

About two-thirds of RCCs present in a local stage, while 16–30% of patients will be diagnosed with metastatic disease [7,11], although pRCCs have a lower risk of presenting in the metastatic stage [74].

The classical triad of presenting symptoms include flank and back pain, hematuria, a palpable flank mass, but is seen in only 15% of cases [75,76], and it correlates with advanced disease and poorer prognosis [77,78]. In one series, hematuria was present in 35% of patients, while 57% were asymptomatic [51]; furthermore, unprovoked hemorrhage can be a presenting manifestation in 8% of cases [79]. Paraneoplastic manifestations, with constitutional signs and symptoms (fever, cachexia, and weight loss), hypertension and/or metabolic anomalies (hypercalcemia, polycythemia, amyloidosis, hepatic dysfunction, and Cushing’s syndrome) can also be present as in other RCCs [76]. For instance, new-onset proteinuria and various neuropathies have been documented specifically in pRCC cases [80,81,82,83].

### 4.2. Prognosis

In total, 39–55% of RCC cases are diagnosed incidentally [53,71,72,73], and the current trend is characterized by a growing detection rate of RCCs featuring a lower stage and grade, especially in older patients [71].

pRCCs usually have a better prognosis than ccRCCs (the main type of RCC), if at the same stage [63,84]. Staging may be the most important prognostic factor [84]. Lower rates of nodal involvement, venous extension and distant metastases when compared to other RCC histotypes, are observed [85,86].

The 2012 WHO/ISUP grading system has prognostic significance [69], and its use is recommended by the latest EAU and American Urological Association (AUA) guidelines [65]. The use of prognostic models encompassing histologic (es. grading, necrosis, and tumor thrombus), staging and clinical (es. ECOG or Karnofsky PS, laboratory parameters, interval from diagnosis to treatment) parameters is strongly recommended by the latest update of these guidelines in localized (the UISS [87], the 2003 [88] or 2018 Leibovich score/model [89], the VENUSS score/model [90], and the GRANT score [91]) and metastatic disease (MSKCC [92] and IMDC [93]) [65,94].

In a recent study on the prognostic significance of the clinical and pathological parameters of 87 pRCCs, higher pT stage and greater size, as well as lymph node metastasis, distant metastasis, and high pathological grade were associated with worse survival. Furthermore, pTNM stage, tumor grade and subtype were potentially related to prognosis for progression-free survival (PFS), but the authors did not find an independent prognostic factor correlated with PFS using multivariate regression models [51].

The latest EAU guidelines do not recommend the use of molecular markers in clinical practice for treatment selection in metastatic pRCCs, even if a prognostic significance for some molecular biomarkers is often present in published studies, mainly because of the lack of external clinical validation [65].

## 5. Diagnosis and Staging

### 5.1. Diagnostic and Staging Algorithms

The first approach to an incidental or clinically manifest solid renal mass is to characterize its appearance using cross-sectional multiphase abdominal imaging since it can be informative regarding the malignant nature and histotype, to begin the clinical staging, and to perform a complete laboratory serum evaluation (complete blood count, serum creatinine, glomerular filtration rate, erythrocyte sedimentation rate, complete liver panel, serum calcium, and coagulation study) and urinalysis. Both the AUA and the EAU guidelines on renal cell carcinoma suggest performing a multiphase abdominal CT or MRI, with emphasis on non-enhanced morphology (lesion size, attenuation, intralesional fat, local invasion, and nodal or abdominal metastasis), enhancement pattern and contralateral kidney morphology [65,94].

The 2017 AUA and 2019 ESMO guidelines recommend expressing RCC stage using the TNM 8th edition [64,94,95]. The 2019 ESMO guidelines for RCC recommend a staging algorithm with contrast-enhanced CT (CECT) of the chest, abdomen and pelvis, or a high-resolution chest CT without contrast administration plus an abdominal MRI in patients with renal insufficiency or allergic to the contrast media. Bone scan and brain CT or MRI are usually performed only if a cerebral metastasis is clinically suspected or in the presence of abnormal laboratory results [64].

The 2017 AUA guidelines recommend using high quality, contrast-enhanced multiphase cross-sectional abdominal imaging (MDCT or MRI) whenever possible, to stage RCCs. Chest imaging techniques can be tailored depending on tumor risk (according to thrombosis, adenopathy, tumor size, infiltrating appearance on imaging, necrosis); therefore, as a consequence, chest radiography may be appropriate for lower risk tumors, whereas a chest CT can be reserved for patients with a remarkable clinical burden or high tumor risk [94].

### 5.2. Imaging Techniques in pRCC

#### 5.2.1. Ultrasound Features

Unenhanced ultrasound (US) is not a first line diagnostic technique in solid RCC diagnosis, but it can be helpful in determining the presence of a solid or cystic pattern of a detected renal lesion that displays an equivocal borderline enhancement or low-absent enhancement despite being hyperdense on CT imaging [96].

pRCCs display a different enhancement pattern from ccRCCs at contrast enhanced ultrasound (CEUS) examination, exhibiting more frequently low enhancement, slow wash-in, fast wash-out patterns; moreover, in >3 cm lesions, enhancement was typically more homogeneous. By combining the three enhancement-features, a positive predictive value (PPV) and a negative predictive value (NPV) of 86.7% and 86.9%, respectively, were reported [97]. However, it was reported that up to 25% of pRCCs can present with an atypical enhancement pattern, similar to that of ccRCCs [98].

Xue et al. reported that CEUS is a helpful tool in distinguishing ccRCCs from non-ccRCCs, but it cannot distinguish pRCCs from chRCCs since both have a pseudocapsule, low enhancement, homogeneous appearance, and fast wash-out [99].

Liang et al. reported that CEUS and quantitative analysis of ROI time-intensity curves can be helpful in distinguishing the three histotypes, especially when combining CEUS with CECT. A total of 54.2% of pRCCs showed inhomogeneous echoes and liquid areas, and most lesions exhibited slow-forward and slow-retrograde with low enhancement [100].

#### 5.2.2. CT Features

PRCC can exhibit low attenuation with non-contrast CT, although <20 Hounsfield units (HU) are usually associated with an underlying benign lesion. Half of pathologically diagnosed pRCCs display <20 HU pre-contrast attenuation [101]. Calcifications are more commonly seen in pRCCs (32%) than in ccRCCs (11%) [102].

On multi-phase CECT, pRCCs show a gradual, progressive enhancement, as in contrast-enhanced MRI [103].

Multidetector-CT with iodine quantification could be used to differentiate pRCCs from ccRCCs. Mileto et al. reported a PPV of 95.8% and a NPV of 93.7% using a 0.9 ng/mL tumor iodine concentration (ITC) threshold, with an area under the curve of 0.923 and a 0.9990 intraclass correlation coefficient among five observers. Furthermore, ITC was also associated with tumor grading [104].

Type 1 and type 2 pRCCs typically have a similar appearance, but a heterogeneous appearance and indistinct margins were observed more frequently in type 2 pRCCs in a case series [105]. Relative excretory phase attenuation was greater for type 2 than for type 1 pRCCs, distinguishing the two subtypes with a 74% PPV and a 71% NPV, in a retrospective study, although these findings need to be validated prospectively [106].

#### 5.2.3. MRI Features

On MR-imaging, pRCCs are typically hyperattenuating lesions [107], with a pseudocapsule that is frequently detected by MRI [108]. These tumors characteristically exhibit a gradual, progressive enhancement after contrast administration, and on T2-weighted images and apparent diffusion coefficient (ADC) map, they exhibit a low signal [107].

A study on MRI with dynamic gadolinium contrast in the corticomedullary phase enhancement (CMP) showed sensitivity of 86%, specificity of 92% in differentiating pRCCs with benign and other malign, except ccRCC [109]. Moreover, Maryellen et al. observed that pRCCs displayed a lesser signal intensity change in the corticomedullary (32.1%) and nephrogenic phases than in ccRCC images (295.6% and 247.1%, respectively). pRCCs had the lowest tumor-to-cortex enhancement indexes at the CMP and nephrogenic phase (NGP), and, using signal intensity changes on CMP to differentiate the two RCC types, the area under ROC curve was 0.99, with a sensitivity of 93% and specificity of 96% using a threshold of 84% [110].

[NO_PRINTED_FORM] Chemical shift MRI can highlight a signal intensity drop on in-phase images versus opposed-phase images, such as an artefact caused by chronic bleeding and hemosiderin deposits in the tumor mass, more frequently in pRCCs than in ccRCCs, a feature that is absent in benign neoplasms [111].

Diffusion-weighted (DW) MR images can help discriminate between RCC subtypes in the preoperative setting since pRCCs exhibit a lower ADC value than other RCCs but that is similar to chRCCs [112].

The use of multiparametric MRI can predict the underlying histology of a renal mass. In a retrospective study of 100 solid renal tumors without macroscopic fat, including double-echo chemical shift, dynamic contrast-enhanced T1- and T2-weighted images, ADC maps were reviewed by two radiologists, allowing to differentiate pRCCs from other renal neoplasms with 37.5% sensitivity and 100% specificity. If prospectively validated, these findings might spare an unnecessary tumor biopsy in hypo-enhanced renal neoplasms with hypointense signal in T2-weighted images [113].

### 5.3. Renal Mass Biopsy

A renal biopsy can be helpful in localized renal masses. The 2019 ESMO guidelines on renal cell carcinoma acknowledge the importance of renal mass biopsy (RMB) using core needle biopsy (CNB) in evaluating the malignant nature of a renal mass. CNB is particularly recommended before ablative procedures (III, B) and before systemic treatment in the metastatic disease stage (III, B).

RMB is recommended also in the case of local disease, with a cortical renal lesion of ≤3 cm in diameter, particularly for frail patients that would undergo local ablation, are at high surgical risk, or who have solitary kidney, impaired renal function, multiple bilateral tumors, or hereditary RCCs.

In patients with small renal mass (SRM) undergoing active surveillance (elderly with comorbidities, short life expectancy and solid tumors of less than 4 cm), RMB is recommended since benign tumors are frequent in this subset of patients (III) [64].

RMB can also be considered in case of a renal mass suspected of being a metastatic lesion due to its imaging appearance and recent history of malignancy with potential renal metastasis, or when an inflammatory or infectious disease is suspected on the basis of suggestive clinical manifestations and a history of prior diagnosis [94].

RMB is not mandatory; it can be avoided in older or frail patients who are candidates for a conservative treatment strategy, regardless of the pathological examination diagnosis [94].

## 6. Treatment

### 6.1. Treatment Strategy for Localized Disease

The main goal of our review is to characterize the diagnostic, prognostic, and therapeutic significance of long non-coding RNA in pap RCC, providing a concise clinical background to contextualize the reported evidence. Therefore, the treatment section will briefly summarize the best therapeutic options according to the latest evidence and guidelines.

Localized disease refers to a stage I to III RCC, without invasion beyond the Gerota’s fascia or distant metastasis, but with a possible extension in the renal or cava vein or invasion of the renal pelvis or calices or perinephric tissues [95]. In this setting, ccRCCs and pRCCs have a similar management strategy, which is based on active surveillance and local ablation or radical resection with a curative goal [65,94]. SRM can be managed with active surveillance or partial nephrectomy (PN) and ablative techniques [114] to spare the residual renal parenchyma and reduce the risk of chronic kidney disease onset and/or progression while achieving favorable oncologic outcomes [115]. PN should be the first choice for cT1a lesions, and nephron-sparing procedures should be considered especially for bilateral tumors, familial RCCs, solitary kidney, and in the presence of proteinuria or chronic kidney disease [94]. PN is feasible also for >4 cm lesions limited to the kidney because of a lower likelihood of tumor recurrence, cancer-specific and all-cause mortality, but it should be used in selected patients because of the higher hemorrhagic risk and likelihood of complications [116]. A recent meta-analysis reported that PN and radical nephrectomy (RN) in patients with pT3a RCC showed similar results, with no differences in cancer-specific mortality, overall survival (OS), cancer-specific survival, relapse-free survival, complications and positive surgical margin, but these findings need to be confirmed by larger prospective studies [117].

Active surveillance for SRM can be chosen above all in the setting of elderly and frail patients with impactful comorbid conditions, or in small tumors (a 2 cm threshold is suggested) [118].

Radical nephrectomy can be curative in localized disease and is usually performed more frequently in tumors with a >T1 stage. It was reported that in patients with T1 lesions, nephron-sparing surgery (NSS) and RN did not affect PFS, regardless of tumor histotype [51]. RN should be considered if tumor size, RMB, and imaging appearance suggest an increased oncologic potential, particularly if CKD, proteinuria, and contralateral kidney abnormalities are absent [94].

Neoadjuvant therapy should be proposed only in the setting of clinical trials. Adjuvant tyrosine kinase inhibitors (TKIs) and immune-checkpoint inhibitors (ICIs) have been investigated, and TKIs have also been approved by the FDA in the adjuvant setting, but similarly to ICIs, they did not improve overall survival after nephrectomy in recent clinical trials [65]. A recent RCT also evaluated non-ccRCC patients (16% of the total population), receiving a 1-year course of adjuvant sorafenib, but it failed to demonstrate a benefit in disease-free survival (DFS) and OS [119]. Although adjuvant Pembrolizumab, an ICI, is now weakly recommended for high recurrence risk ccRCCs by the EAU guidelines, there is no evidence on its use in pRCCs [65,120].

In summary, adjuvant therapy is not an option for pRCCs [64,65].

### 6.2. Treatment Strategy for Advanced or Metastatic Disease

Cytoreductive nephrectomy (CN) was evaluated in modern studies in ccRCC patients who were risk-stratified using the MSKCC [121,122]. In intermediate- and poor-risk asymptomatic patients, upfront CN is associated with morbidity and mortality and is no longer the first-choice treatment, but it can be considered if local symptoms or near-complete response to systemic treatment is present [64]. Immediate CN can be performed in patients with a good performance status not requiring systemic therapy or with oligometastases that can be completely excised [65].

In metastatic disease, systemic target therapy is of paramount importance. Recent research has focused on VEGF-R inhibitors and ICIs. The therapeutic algorithm varies with regard to ccRCCs and other non-ccRCCs; furthermore, high-quality data from phase III trials are still missing because this subset of patients is usually excluded from large trials [64], Such patients should be referred to a clinical trial whenever possible and appropriate [65].

A recent study investigated the oncological outcomes in metastatic non-ccRCC patients receiving the TKI sunitinib and the mTORi everolimus and reported that sunitinib improved PFS versus severolimus [123]. Another TKI, cabozantinib, was recently investigated and showed remarkable efficacy and tolerability results [124,125]. This led to a comparative study between different VEGF-R inhibitors which showed that cabozantinib resulted in a benefit in PFS and response rate versus sunitinib, whereas savolitinib and crizotinib did not achieve this outcome [126]. In a multicenter, phase 3, open-label RCT involving 60 patients with MET-mutated pRCCs, savolitinib was associated with OS, PFS and objective response rate (ORR) benefit, with fewer adverse events (≥3 grade) versus sunitinib [127].

The 2019 ESMO guidelines state that an acceptable option for pRCCs is the use of cMET inhibitors if a cMET mutation or amplification is demonstrated, suggesting a near-future clinical implication for some biomarkers [64].

The latest update of the EAU guidelines on RCCs now weakly recommend offering cabozantinib to patients with advanced pRCCs without performing molecular testing, while savolitinib is recommended for MET-driven tumors [65].

Pembrolizumab and nivolumab are two ICIs that were recently investigated, even in a non-ccRCC context, and are currently an option for advanced pRCCs, as are TKIs.

The combination of these two ICIS is used for ccRCCs and is a possible option for pRCC. A retrospective study of 18 non-ccRCC patients reported partial response in 33% and stable disease in 16.7% of cases, and a median PFS of 7.1 months; however, around two-thirds of patients experienced treatment-related adverse events [128]. A phase 3b/4 trial on 52 non-ccRCC patients reported a 20% objective response rate (n = 46) and similar safety profile but above all, the responses were all achieved in the papillary or unclassified subgroups, suggesting that pRCCs are particularly ICI sensitive [129]. Single-line therapy with pembrolizumab is also an option and is weakly recommended by the EAU guidelines [65] since its use resulted in long-term OS in the pRCCs in a phase II trial [130].

Nivolumab is another valuable option, and a phase IIIb/IV trial documented a 14% overall response rate in 44 non-ccRCCs, with 24 cases of pRCCs, a median PFS of 2 months and OS of 16 months [131].

## 7. Long Non-Coding RNA in pRCC

### 7.1. The ceRNA Network

Non-coding RNAs (ncRNAs) are functional elements which have critical roles in gene expression, at the epigenetic, transcriptional, post-transcriptional levels [132]. NcRNA and messenger RNA (mRNA) form intricate gene expression regulatory networks called competitive endogenous RNA (ceRNA) networks [133,134]. In the ceRNA hypothesis, lncRNAs can bind micro RNAS (miRNAs), resulting in the modulation of the mRNA levels targeted by the sponged miRNA. In general, miRNAs silence a target gene by binding to the 3′-untranslated region (3′-UTR) of the target mRNA. LncRNAs contain miRNA response elements that competitively bind miRNAs interacting with other RNA transcripts containing MREs. This leads to additional regulation of target gene expression and increased complexity in the biological processes [135]. So far, various forms of evidence indicate that this regulatory network plays an essential role in the occurrence, development, and regulation of tumors, including the pRCC [134]. The Cancer Genome Atlas (TCGA) is a great source of data to construct a ceRNA network in different types of cancers. He et al. analyzed 1251 lncRNA–miRNA–mRNA interactions, selecting eight differentially expressed genes (IGF2BP3, PLK1, LINC00200, NCAPG, CENPF, miR-217, GAS6-As1, and LRRC4) as prognostic signature. All these genes have been studied for their association with tumor progression, as they are implicated in the regulation of cell growth, migration and response to drug [134]. IGF2BP3, PLK1, NCAPG, CENPF, and miR-217 were increased in late-stage pRCC and were considered predictive for poor survival and high mortality in patients. Conversely, pRCC patients within the high GAS6-AS1 had better overall survival, and GAS6-AS1 and LRRC4 both interacted with miR182: by acting as a ceRNAs, the lncRNA GAS6-AS1 can modulate the levels of the LRRC4 in pRCC. Consistent with this hypothesis, the pRCC patients with high expression of LRRC4 were also associated with higher overall survival, suggesting its role as a tumor suppressor [134]. In another study, differential expression profile analysis of TCGA database identified 1970 mRNAs, 1201 lncRNAs and 96 miRNAs as genes with significantly different expression between pRCC and paracancerous tissues [136]. The evaluation of the ceRNA network highlighted the lncRNAs MEG3 (maternally expressed 3) and PWRN1 (Prader – Willi region non-protein coding RNA 1), miRNAs miR-508, miR-21 and miR-519 as important genes. The downregulation of MEG3 expression levels was validated in a cohort of 12 pRCC tumors and adjacent non-tumor tissues [136]. MEG3 has been reported to act as a lncRNA tumor suppressor in various tumors via interactions with p53 and the regulation of the expression of its target genes [137]. In addition, the pan-cancer analysis of TGCA suggested that the expression of the G protein regulated inducer of neurite outgrowth 1 (GPRIN1) increased in 16 tumor types, included pRCC, and that it is correlated with poor prognosis [138]. Based to the regulation mechanism of ceRNA, lncRNA should be expected to be negatively correlated with miRNA, but positively correlated with GPRIN1. Along this line, the ceRNA network was reconstructed by detecting the downregulation of miR-140-3p and the increase in the lncRNAs, LINC00894, MMP25-AS1 and SNHG1. Thus, the latter might be upstream the miR140-3p/GPRIN1 axis in pRCC [138]. A complementary approach was investigated by Jia et al. [139], who focused the analysis on the core effector molecules involved in tumor microenvironment modulation of pRCC. By analyzing a cohort of 233 pRCC patients, whose expression data were available in the TCGA database, they determined a stromal score and divided the patients according to high and low scores. From the analysis of differentially expressed genes between the two groups, the lncRNA GUSBP11/miR-432-5p/CAMK2B axis was selected as a promising ceRNA network. Data validation in tissue samples confirmed that levels of CAMK2B and GUSBP11 were lower, while those of miR-432-5p were higher in tumor tissue than in paired tumor-adjacent tissue, suggesting a protective role of the axis in pRCC. The cellular mechanism of action for CAMK2B relates to inhibiting proliferation and remodeling angiogenesis and fibrogenesis, likely through inhibiting downstream effector genes, including vascular endothelial growth factor (VEGF) and transforming growth factor (TGF)β, and close homolog of L1 (CHL1) [139]. To investigate the regulatory potential of pRCC toward the infiltrating immune cells related to tumorigenesis, the evaluation of the ceRNA network was based on the 318 samples from TCGA, including 285 pRCC and 33 normal control samples [140]. MiR-29c-3p (miRNA), COL1A1 (protein-coding RNA), and H19 (lncRNA) were significantly correlated. Furthermore, COL1A1 was negatively associated with M1 macrophage infiltration and positively related with infiltrating M2 macrophage, while H19 was positively linked with M2 macrophage infiltration, suggesting that the crosstalk between the H19-miR-29c-3p-COL1A1 axis and the infiltration of M1 and M2 macrophages can regulate pRCC development. Such axis might promote the polarization of M2 macrophages and inhibit M1 macrophage activation through the Wnt signaling pathway, co-operating in the pRCC tumorigenesis and leading to poor overall survival of pRCC patients [140].

Most of the abovementioned lncRNAs can be also found in the ceRNA network of other cancers, such as LINC00200 (liver [141] and gastric [142] cancers), GAS6-AS1 (lung [143], breast [144] and liver cancer [145]), MEG3 (breast [146] and bladder [147] among others), PWRN1 (gastric [148]), LINC00894 (breast cancer [149]), SNHG1 (pancreatic [150], liver [151], and breast [152]), and H19 (lung [153], thyroid [154], and gallbladder [155]), suggesting general mechanisms of tumor development.

However, lncRNAs across tumors do not share the same sponged miRNAs, indicating a great complexity of these systems. LncRNAs are able to interact simultaneously with multiple targets and so affect plenty of cellular pathways, creating a kind of cascading effect, which drives the tumor growth. These mechanisms may contribute to a better under-standing of the pathogenesis and provides potential therapeutic strategy as well as diagnostic/prognostic biomarkers.

### 7.2. Prognostic Long Non-Coding RNA in TCGA Dataset

As mentioned above, there were several studies investigating the prognostic value of lncRNAs and exploring ceRNA networks in pRCC by using the cancer genome atlas (TCGA) datasets.

Chen et al., by using the TCGA datasets, found a four immune-related lncRNAs prognostic signature able to predict the OS time outcomes of patients with pRCC [156]. Indeed, the expression levels of these four lncRNAs (AC015922.3, AL031710.1, AC099850.3, and LIFR-AS1) was significantly associated with OS of pRCC: a lower expression of AC015922.3, AL031710.1, and LIFR-AS1 and a higher expression of AC099850.3 were associated with a worse survival rate (p value of 0.025, 0.002, 0.001, and <0.001, respectively, with KM survival analysis) [156]. The authors validated the four-lncRNA prognostic model in an independent cohort of 66 pRCC patients, where the expression levels of the lncRNAs were analyzed by RT-qPCR [156]. By using multivariate analysis, they also showed that the four-lncRNAs prognostic model was an independent predictor of OS in pRCC patients (hazard ratio, 4.49; 95% CI, 1.26–16.06) [156]. In addition, by gene set enrichment analysis (GSEA), they also showed that the high-risk group of patients had a significant immune response, as expected [156].

The lncRNA-sequencing data of pRCC patients in TCGA was analyzed also by Lan et al. [157]. They found 780 differentially expressed lncRNAs between pRCC tissue and normal kidney tissues [157]. By univariate Cox proportional hazards regression, they showed that 37 differentially expressed lncRNAs displayed remarkable prognostic value [157]. Then, by using the multivariate cox regression analysis, they constructed a prognosis index that consisted of seven lncRNAs (including AFAP1-AS1, GAS6-AS1, RP11-1C8.7, RP11-21L19.1, RP11-503C24.1, RP11-536I6.2, and RP11-63A11.1) [157]. This prognosis index displayed considerable predictive potential for disease progression and could be an independent prognostic indicator for pRCC patients [157]. High-risk and low-risk patients, classified based on the prognosis index, showed different sets of differentially expressed genes in tumor tissues compared to normal tissue [157]. Based on these differentially expressed genes, with gene set enrichment analysis, the authors found different signaling pathways that could help clarify the molecular mechanisms underlying the different outcomes of high-risk compared to low-risk patients [157]. Indeed, a total of 156 pathways were considerably enriched in the high-risk group, including KEGG_VASCULAR_SMOOTH_MUSCLE_CONTRACTION, KEGG_TGF_BETA_SIGNALING_PATHWAY, and KEGG_MAPK_SIGNALING_PATHWAY [157]. On the other hand, 21 pathways were enriched in the low-risk group, including some cancer-related pathways such as KEGG_OXIDATIVE_PHOSPHORYLATION, and KEGG_REGULATION_OF_AUTOPHAGY [157]. However, the findings obtained by this computational analysis required experimental verification.

In another study, Liu et al. investigated the potential of immune-related lncRNAs (IR-lncRNAs) on predicting tumor progression and prognosis in pRCC patients [158]. By using the TCGA dataset, the authors first calculated immune scores based on the expression level of immune related genes, then identified the most relevant IR-lncRNAs by Pearson correlation analysis of immune score and the lncRNA expression level [158]. By integrating the expression profiles of lncRNA and overall survival (OS) in the 322 pRCC patients, by COX regression analysis, they identified IR-lncRNAs that were significantly correlated with the OS of pRCC patients. Then, the authors used the four IR-lncRNAs (AP001267.3, AC026471.3, SNHG16 and ADAMTS9-AS1) with the most remarkable prognostic values to establish an immune-related risk score (IRRS) model able to distinguish high risk patients from low-risk patients [158]. The expression levels of AC026471.3 and SNHG16 were increased, while those of AP001267.3 and ADAMTS9-AS1 were decreased with the increase in risk score. PCA analysis revealed significantly different distributions between high-risk patients and low-risk patients, and the GSEA analysis showed also a different immune status between the two groups of patients. IRRS resulted as an independent prognostic factor with an AUC of risk score of 0.958 and correlated with the OS of pRCC patients [158].

In another study, Yan et al., by using lncRNAtor datasets, demonstrated that 137 lncRNAs were specifically dysregulated (105 lncRNA upregulated and 32 down regulated) in pRCC with respect to normal kidney tissue and different from that dysregulated in other tumors, such as bladder cancer, clear cell RCC, and prostate adenocarcinoma [159]. On the other hand, 34 lncRNAs were differently expressed in all four urologic cancer types [159]. Then, the authors conducted a coexpression network analysis, evaluating the correlation existing in differentially expressed mRNAs and lncRNAs, to forecast common differently expressed lncRNA functions. They found 15 lncRNAs (RPL32P3, RP11-66N24.3, SNHG1, SNHG11, HERC2P2, AC005154.5, RP5-1180C10.2, RP11-65F13.2, GAS5, AL589743.2, MIR22HG, FDG5-AS1, ZNFX1-AS1, WDFY3-AS2, and RP11-57H14.4) and 498 mRNAs in pRCC. RPL32P3, RP11-66N24.3, RP5-1180C10.2, and SNHG11 were identified to be key regulators in the progression of urologic cancers [159]. Then, the authors used the set of coexpressed mRNAs to determine the role of common differently expressed lncRNAs by using gene ontology (GO) and Kyoto Encyclopedia of Genes and Genomes (KEGG) pathway analyses. They showed that common differently expressed lncRNAs were involved in biological process such as regulating translation, translational initiation, rRNA processing, proteasome-mediated ubiquitin, and DNA replication in pRCC. KEGG analysis showed that common differently expressed lncRNAs were enriched in ribosome, spliceosome, proteasome, Fanconi anemia pathway, and biosynthesis of amino acids in pRCC [159]. The authors also constructed cancer-specific lncRNAs coexpressing networks in the four urologic cancer types under investigation and found that 49 lncRNAs and 794 mRNAs were in pRCC. The following pRCC-specific lncRNAs were identified to be key regulators in the progression of urologic cancers: RP11-510M2.2, ZNF252P-AS1, UBE2Q2P2, ADORA2A-AS1, RP11-279F6.1, and MRPL23-AS1 in pRCC. Through GO analysis, the authors found that pRCC-specific lncRNAs were primarily involved in the morphogenesis of cilium, the process of metabolic, oxidation of fatty acid beta, homeostasis of lipid, and fatty acid beta-oxidation, using acyl-CoA dehydrogenase. KEGG analysis suggested that pRCC-specific lncRNAs principally took part in metabolic pathways, staphylococcus aureus infection, biosynthesis of antibiotics, valine, leucine, and isoleucine degradation, and carbon metabolism [159]. The dysregulation of DLGAP1-AS3, SPON1, ULK4P3, RPL34-AS1, RP11-557H15.3, RP11-368J21.3, ANKRD18DP, LINC00607, and ADORA2A-AS1 were significantly correlated to OS in pRCC and could thus serve as prognostic markers [159].

In addition, PVT1 was overexpressed in all four urologic cancers, and highly expressed PVT1 was negatively correlated with overall survival time [159]. It was shown that PVT1 could sponge miRNAs and bind proteins to modulate cell proliferation and invasion. Indeed, PVT1-214 was reported to promote the proliferation and invasion of colorectal cancer by stabilizing Lin28 and interacting with miR-128 [160]. In hepatocellular carcinoma, PVT1 promoted cell proliferation and inhibited apoptosis by recruiting EZH2, stabilizing MDM2 protein expression and restraining P53 expression [161].

So, this study reveals the prognostic value of common and cancer-specific differently expressed lncRNAs in pRCC outcomes, together with other urological cancer [159]. All these findings, however, need further and in-depth studies and experimental validation.

In another study, Jia et al. explored how the tumor microenvironment (TME) and in particular, the TME effector genes and their competitive endogenous RNA (ceRNA) networks (previously described in Section 7.1 of this review) affect pRCC tumor progression [139]. Analyzing gene transcript, miRNA, and lncRNA expression data of pRCC on the TCGA datasets, they performed estimation of stromal scores and immune scores and found that high stromal scores were associated with a poor prognosis in pRCC. They then identified 2509 differentially expressed genes in high stromal score tumors compared to low stromal score tumors, including 1668 mRNAs, 783 lncRNAs, and 58 miRNAs [139]. Through weighted gene co-expression network analysis (WGCNA), they identified the competitive endogenous RNA (ceRNA) network lncRNA GUSBP11/miR-432-5p/CAMK2B as a promising prognostic factor. The authors also showed that CAMK2B promotes stromal TME remodulation and inhibits proliferation in pRCC [139]. So, the stromal score related network lncRNA GUSBP11/miR-432-5p/CAMK2B showed prognostic potential in pRCC, and CAMK2B may represent an effective therapeutic target [139].

Another interesting study investigated new possible lncRNAs in the pRCC scenario as prognostic biomarkers [162]. Using the TCGA data, a total of 14,447 lncRNAs were extracted from the database, and 8044 lncRNAs were identified as expressed genes. Among them, 1001 lncRNAs were differentially expressed in pRCC tissues with respect to the healthy counterpart. In particular, 546 were overexpressed, while 455 were downregulated. The authors performed the univariate Cox, and the LASSO regression analyses in order to highlight the most prominent lncRNAs. This statistical approach underlined about 17 key lncRNAs, which were considered to create a prognostic model. Based on the multivariate Cox regression, six lncRNAs were identified as potential prognostic biomarkers. Subsequently, Kaplan Meyer survival analysis validated the values of five lncRNAs, AC024022.1, AC087379.2, AL352984.1, AL499627.1, and GAS6-AS1. The lncRNA map underlined that lnc-RNA-AC024022.1 (ENSG00000250781) was the sequence of ‘Homo sapiens BAC clone RP11-63A11’; lnc-RNA-AC087379.2 (ENSG00000254695) was defined as ‘Homo sapiens chromosome 11, clone RP11-396020; and lnc-RNA-AL352984.1 (ENSG00000258386) displayed ‘Homo sapiens BAC R-187E13 of library RPCI-11 from chromosome 14. The lncRNA AL499627.1 (ENSG00000260542) was derived from clone RP13-379O24 on human chromosome 20 whereas lnc-RNA-GAS6-AS1 (ENSG00000233695) was in line with Homo sapiens GAS6 antisense RNA-1. Among these five pivotal lncRNAs, the lncGAS6-AS1 had been already investigated in the small cell lung cancer, where its downregulation was linked to a poor prognosis [163]. In line with the literature, also in this work, the reduced levels of lncRNA correlated with a worse OS rate, suggesting a possible protective role of this lncRNA in the prognosis of pRCC. At the same time, also the decreased expression of lncRNA AC024022.1 was related to a poor prognosis of pRCC in the statistical analysis; on the contrary, low levels of lncRNA AC087379.2, lncRNA AL352984.1 and AL499627.1 correlated with a better OR rate, indicating a plausible tumorigenic action.

Finally, the authors decided to explore whether the selected five lncRNAs were located in the cytoplasm or in the cytosol, in order to hypothesize a role as ceRNA. Using lnc-Locator based on the sequences acquired from lncRNAMap, the authors found that only lnc-RNA-AC024022.1(ENSG00000250781) and lncRNA AC087379.2 (ENSG00000254695) were located in the cytoplasm, whereas GAS6-AS1 (ENSG00000233695) were likely in the cytosol. Therefore, it is plausible that these three lnc-RNA probably acted as ceRNA, playing an important role in modulating the stability or translation of mRNA.

Another recent work, always starting from the TCGA database, investigated new possible prognostic lncRNAs [164]. The authors analyzed with univariate COX regression the expression profile of 367 cancer-related lncRNAs with overall survival data and identified 26 lncRNAs as possible prognostic biomarkers. Then, they conduced multivariate COX regression to create a risk model of prognosis. A total of 16 lncRNAs were eliminated, and the remnant ones (RP11-573D15.8, LINC01317, RNF144A-AS1, TFAP2A-AS1, LINC00702, GAS6-AS1, RP11-400K9.4, LUCAT1, RP11-63A11.1, and RP11-156L14.1) were included in the model. The discovered signatures alone were able to stratify patients at high risk of poor survival, independently from the other clinical variables. This risk-score prediction model showed AUC values at 1, 3, and 5 years over than 0.75, indicating that the 10 cancer-related lncRNA signature is a good predictor of prognosis in pRCC patients. Among the selected lncRNAs, RNF144A-AS1, called GRASLND, was indicated to play a crucial regulatory role in stem cell chondrogenesis [165], and it can promote the migration an invasion of bladder cancer cell [166]. The other identified lncRNA, the LINC00702, was involved in several type of cancers influencing the progression of the tumor through molecular pathways, such as the Wnt/β-catenin pathway and PTEN/PI3K-AKT pathway [167]. Additionally, LUCAT1 promotes tumorigenesis in different oncological assets together with cancer therapy resistance, particularly in osteosarcoma and non-small-cell-lung carcinoma [168]. All these three lncRNAs were highlighted for the first time for pRCC by the authors.

On the other hand, as well described in the literature [157], RP11-63A11.1 was already investigated in the pRCC universe, but in this article, the authors found an important association with the clinicopathological variables and with the prognosis of pRCC patients. In fact, the augmented expression of RP11-63A11.1 inhibited the proliferation and promoted the apoptosis of pRCC cells, suggesting a possible role as a tumor suppressor. Regarding the other new elucidated long non-coding RNAs, including RP11-573D15.8, LINC01317, RP11-400K9.4, and RP11-156L14.1, the authors decided to combine them with the well-known above-mentioned lncRNAs, creating an innovative robust tool able to predict the overall survival in the pRCC scenario. Finally, in order to investigate the biological functions of these selected lncRNAs, a WGCNA analysis was performed to detect which gene modules could be associated with the discovered signatures. In particular, two genes involved in cell division, proliferation and cell cycle were elucidated, suggesting that pRCC with high-risk score propagates more than pRCC with low-risk score.

Zhu et al., in a recent original article [169], investigated the expression profiles of lncRNAs in a cohort of 289 pRCC samples and 32 normal renal tissues from the TGCA database. The authors underlined about 1928 differentially expressed mRNAs, 981 differentially expressed lncRNAs and 52 differentially expressed miRNAs between pRCC samples and normal renal tissues. Subsequently, the top 200 differentially expressed mRNA were selected to perform GO and KEGG pathway analyses. Of interest, these differentially expressed mRNAs were enriched in some carcinogenesis related pathways, including the Wnt signaling pathway and arachidonic acid metabolism. Then, with univariate and multivariate Cox regression analysis, a 3-mRNA prognostic signature was established, including ERG, RRM2, EGF. ERG is a well-known oncogene, involving in hematopoiesis, chondrocyte maturation, bone development, apoptosis and cell migration [170]. RRM2 represents an essential enzyme involved in the DNA replication and repair [171] and its overexpression is related to the establishment of advanced form of bladder cancer, head and neck cancer, adrenocortical cancer, breast cancer and pancreas adenocarcinoma [172]. EGF is involved in the epidermal growth factor family and plays a crucial role in the proliferation and differentiation of cells [173]. Therefore, EGF alteration results in carcinogenesis development [174]. A higher expression of these three mRNAs was linked to a worse prognosis of pRCC. The 3-mRNA signature showed a high prognostic power, with AUC of 0.815, while using a median risk score as the cutoff. Dividing patients according to high and low risk based on this signature, it was shown that the 5-year overall survival rates were 65.5% (95%CI: 54.9–78.1%) in the high-risk group and 84.5% (95%CI: 74.8–95.4%) in the low-risk group, respectively. Then, the authors constructed a lncRNA–miRNA–mRNA network consisting of 11 miRNAs, 28 mRNAs, and 57 lncRNAs based on the combination of the lncRNA–miRNA and miRNA–mRNA interactions. By using the Kaplan–Meier method in order to evaluate the associations between the key members of ceRNA network and prognosis of pRCC patients, the authors found 12 of 57 differentially expressed lncRNAs associated with the prognosis of pRCC. Six of them, including AP000525.1, DNM3OS, GDNF-AS1, GLIS3-AS1, LINC00310, and LINC00462, were positively connected with OS of pRCC, while the other six lncRNAs, including COL18A1-AS1, CRNDE, GAS6-AS1, GPC5-AS1, LINC00327, and SACS-AS1, were negatively correlated with OS [169].

To sum up, the abovementioned works, focused on TGCA data, deeply investigated the differentially expressed lncRNAs in pRCC, highlighting a strong relationship between some lncRNAs levels and the oncological prognosis of patients affected by this type of renal cancer. In fact, the selected lncRNAs are often involved in crucial biological pathway related to carcinogenesis and metastatic spreading. Therefore, the altered expression of these lncRNAs was shown to correlate positively or negatively with pRCC patients’ survival.

In our point of view, the results emerging from these studies could be a strong indication of what also clinicians could perform on the tumoral renal tissue of patients undergoing radical or partial nephrectomy. It is intuitive to hypothesize a new possible medical algorithm, where the pathologists could improve their histo-pathological tissue examinations with a new molecular analysis related to differentially expressed lncRNAs, with the aim to define risk stratification of poor prognosis for this kind of patient. The authors therefore believe that an innovative personalized strategy like this could easily enter the daily clinical practice, significantly ameliorating both the lifespan of patients and the medical decisions in term of targeted treatment. Of course, all these findings require a clinical validation in large cohort of patients before entering clinical practice.

### 7.3. Prognostic Value of Ferroptosis Related lncRNAs in pRCC Patients

The regulated cell death (RCD) pathways play a pivotal role in organism growth and homeostasis and as well are implicated in pathogenesis and treatment resistance in numerous clinical conditions, including cancer. For many years, the caspase-dependent apoptosis has been considered the unique form of RCD; conversely the promotion of this mechanism represented one of the principal therapeutic targets in oncology. However, cancer cells can develop numerous mechanisms that drive resistance to apoptosis, raising the necessity to identify alternative target in promoting cell death and drug resistance in oncology. In the last decade, more than 20 mechanisms of RCD alternatives to apoptosis have been discovered [175] and some of them have been reported to be active in cancer; the most important include entosis, necroptosis, pyroptosis, and ferroptosis [176]. Ferroptosis, unlike apoptosis and autophagy, is a recently discovered RCD mechanism characterized by intracellular accumulation of iron- and reactive oxygen species (ROS) inducing harm, eventually leading to cell death due to lipid peroxidation of cell membranes [177]. Ferroptosis is involved in many diseases and clinical conditions, including neurodegeneration, stroke, ischemia reperfusion and tumors [178]. With respect to what happen in other pathological processes, cancer cells have an increased iron avidity due to the necessity of rapidly proliferating and are dramatically more sensible to iron depletion; on the other hand, increasing the intracellular iron promotes tumor cell death through induction of ferroptosis [179].

NcRNAs are increasingly considered fundamental regulators of ferroptosis though direct and indirect mechanisms, including blocking of iron intake, regulating ROS productions, decreasing antioxidant capacity, increasing intracellular ferrous iron and regulating ferroptosis sensitivity [180]. Recently, some researchers have focused their attention to the regulation activity of lncRNA. In fact, in lung cancer, the silence of lncRNA ZFAS1 suppresses ferroptosis, reducing inflammation and lipid peroxidation [181]. Renal cell carcinomas are strongly sensible to ferroptosis due to the high dependance from glutathione peroxidase (GPX4) in order to prevent ROS accumulation and cell death; for these reasons, targeting these pathways may overcome treatment resistance. Furthermore, identifying and dosing lncRNAs involved in ferroptosis can promote the development of new biomarkers for diagnosis and prognosis of many neoplasms. At the present moment, few studies are available for exploring the role of ferroptosis in the diagnosis and prognosis of pRCC and are particularly focused on the lncRNA signature associated with overall patient survival.

Recently, Dang and colleagues constructed a prognostic signature of ferroptosis related lncRNAs in patients affected by pRCC. A total of 285 cancer tissue and 32 adjacent normal tissue from pRCC patients were included; the data from RNA sequencing and clinical information were downloaded from the TCGA. A total of 15 ferroptosis-related lncRNAs (ZFAS1, AC010624.2, AL031710.1, AL355102.4, MNX1-AS1, AC109460.1, AC127537.1, AC099850.4, LINC02154, AC024022.1, AC026401.3, LINC02535, ADAMTS9-AS1, AC107464.2, and MIR4435-2HG) were selected by univariate and multivariate Cox analyses, performed to evaluate the relationship between the expression of ferroptosis-related lncRNAs and patients’ OS. By using these lncRNAs, the authors constructed a ferroptosis-related lncRNA prognosis model to predict the prognosis of patients with pRCC, multiplying each ferroptosis related lncRNA expression with a specific coefficient. The patients were then classified as low risk (overall risk score below the median value) or high risk (overall risk score over the median value). Univariate and multivariate Cox analyses revealed that the lncRNA signature was an independent prognostic factor of OS of patients with pRCC (HR:1.005, 95CI: 1.002–1.007). The high-risk group had a higher level of immune cell infiltration compared with the low-risk group, as well as a significantly increased incidence of inflammation and activation of type I interferon pathway. The authors also showed that MNX1-AS1, ZFAS1, MIR4435-2HG, and ADAMTS9-AS1 were significantly correlated with the sensitivity of some chemotherapy drugs in NCI-60 cell lines. Taken together, these results suggest a potential prognostic value of this ferroptosis-related lncRNAs signature, which can be incorporated in a monogram with clinicopathological characteristics for the clinical management of patients with pRCC [182].

Similarly, based on the information of patients with pRCC from TCGA, Tang and colleagues developed a risk factor model using differentially expressed ferroptosis-related lncRNAs in pRCC patients. They identified a five lncRNAs prognostic signature composed of AC099850.3, LINC02535, LNCTAM34A, LINC00462, and FOXD2-AS1, calculating risk scores for samples of pRCC from TCGA and choosing the median as the cutoff. The high-risk pRCC group of patients had a shorter survival time compared to that of the low-risk pRCC group (*p* < 0.001). In addition, the AUC values of 1-, 3-, 5-year survival rates for the constructed signature were 0.908, 0.884, and 0.821, respectively, suggesting a high predictive value in pRCC. Univariate and multivariate Cox regression analyses further revealed that the risk model was an independent factor with prognostic value for predicting the OS of patients with pRCC [183].

Figure 1 summarizes the lncRNAs that were shown to have a prognostic value in pRCC patients, also highlighting common players identified from the authors of the studies described in this review article. GAS6-AS1 seems to be the most promising, as it was confirmed as a prognostic biomarker in five different studies.

## 8. Conclusions

The expression of many lncRNAs is altered in pRCC. LncRNAs, together with miRNAs and mRNAs, form intricate gene expression regulatory networks that play a crucial role in the occurrence, development, and regulation of cancers, including pRCC. LncRNAs show promise as new prognostic biomarkers of pRCC able to improve the management of patients and may be attractive novel therapeutic targets, guiding personalized cancer treatment.

## Figures and Tables

**Figure 1 cells-11-01658-f001:**
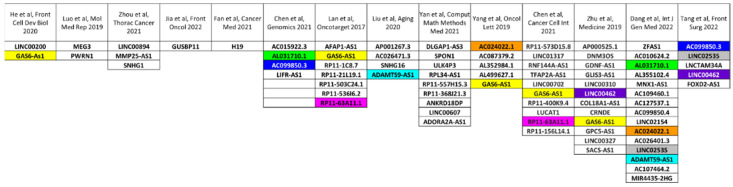
Prognostic lncRNA in pRCC patients, derived from refs [134,136,138,139,140,156,157,158,159,162,164,169,182,183].

**Table 1 cells-11-01658-t001:** 2016 WHO classification of kidney tumors, focus on the subclassification of renal cell tumors [28].

Renal Cell Tumors	Mixed Epithelial and Stromal Tumor Family
Clear cell renal cell carcinoma	
*Multilocular cystic renal neoplasm of low malignant potential*	**Metanephric tumors**
*Papillary renal cell carcinoma*	
*Hereditary leiomyomatosis and renal cell carcinoma-associated renal cell carcinoma*	**Nephroblastic and cystic tumors occurring mainly in children**
*Chromophobe renal cell carcinoma*	
*Collecting duct carcinoma*	**Mesenchymal tumors**
*Renal medullary carcinoma*	
*MiT family translocation renal cell carcinomas*	**Mesenchymal tumors occurring mainly in children**
*Succinate dehydrogenase-deficient renal carcinoma*	
*Mucinous tubular and spindle cell carcinoma*	**Mesenchymal tumors occurring mainly in adults**
*Tubulocystic renal cell carcinoma*	
*Tubulocystic disease-associated renal cell carcinoma*	**Neuroendocrine tumors**
*Clear cell papillary renal cell carcinoma*	
*Renal cell carcinoma, unclassified*	**Miscellaneous tumors**
*Papillary adenoma*	
*Oncocytoma*	**Metastatic tumors**

**Table 2 cells-11-01658-t002:** The WHO/ISUP grading system for pRCC and ccRCC, adapted from [38].

Grade	Criteria from the Original Classification for pRCC and ccRCC
Grade 1	Tumor cell nucleoli absent or inconspicuous and basophilic at 400× magnification
Grade 2	Tumor cell nucleoli conspicuous and eosinophilic at 400× magnification and visible but not prominent at 100× magnification
Grade 3	Tumor cell nucleoli conspicuous and eosinophilic at 100× magnification
Grade 4	Tumors showing extreme nuclear pleomorphism, tumor giant cells and/or the presence of any proportion of tumor showing sarcomatoid and/or rhabdoid differentiation

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
