# Peer review of "Long Non-Coding RNAs as Novel Biomarkers in the Clinical Management of Papillary Renal Cell Carcinoma Patients: A Promise or a Pledge?"

_cells, 2022, doi:10.3390/cells11101658_

Round 1

Reviewer 1 Report

In this manuscript, Trevisani et al. provide a review of the recent progress of long non-coding RNA studies in papillary renal cell carcinoma. Papillary Renal Cell Carcinoma (pRCC) represents the second most common subtype of renal cell carcinoma, and previously was not well studied because of its rarity. However, more and more studies on it have gained attention to it, and a review is in demand to summarize these advances. This review is comprehensive in many aspects and is of interest to the audience of Cells. I have a few comments for the authors:

1, I suggest a separate section discussing the genetics of pRCC. As mentioned, MET gene mutation may increase the genetic risk of pRCC, while this part is currently under “3.3. Macroscopic and microscopic anatomy”, which I don’t think is a suitable place to fit in. Since genetics is an essential part of understanding pRCC, it is better to discuss this separately.

2, In section 7.1, the authors reviewed articles on understanding lncRNAs and their roles in pRCC, focusing on the ceRNA network. Since many studies have been involved, have the authors compared them thoroughly and identified common and different ceRNA networks within them? Please elaborate more on the consistency and discrepancy of the current studies.

3, In section 7.2, the authors discussed several papers on the prognostic analysis of lncRNAs and OS. Many studies claimed hundreds of lncRNAs have been altered in cancer patients. How would this information be helpful to the treatment of the disease? What would be strategies to use these studies to guide clinical treatment? I look forward to more discussion about this since it is important to transform the knowledge from basic research to clinical application.

4, Please pay attention to the spelling of some words and make sure they are in modern English. For example, in the abstract, ‘whith’ should be ‘with’, and “hystological” should be “histological”, etc.

The inclusion of clinical features and treatment strategies in this review is very helpful for understanding the pathology of the disease. With a combination of clinical views and molecular signatures, this review would be beneficial to the scientists and physicians who study pRCC. I suggest a minor revision, and I look forward to seeing the revised manuscript.

Author Response

Answers to Reviewer 1

In this manuscript, Trevisani et al. provide a review of the recent progress of long non-coding RNA studies in papillary renal cell carcinoma. Papillary Renal Cell Carcinoma (pRCC) represents the second most common subtype of renal cell carcinoma, and previously was not well studied because of its rarity. However, more and more studies on it have gained attention to it, and a review is in demand to summarize these advances. This review is comprehensive in many aspects and is of interest to the audience of Cells. I have a few comments for the authors:

1, I suggest a separate section discussing the genetics of pRCC. As mentioned, MET gene mutation may increase the genetic risk of pRCC, while this part is currently under “3.3. Macroscopic and microscopic anatomy”, which I don’t think is a suitable place to fit in. Since genetics is an essential part of understanding pRCC, it is better to discuss this separately.

Thanks for your suggestion. We provided an individual subsection named “3.3. Molecular and genetic features”, in which we focused the discussion around pRCC genetics, with some minor changes and a new reference (ref 50).

2, In section 7.1, the authors reviewed articles on understanding lncRNAs and their roles in pRCC, focusing on the ceRNA network. Since many studies have been involved, have the authors compared them thoroughly and identified common and different ceRNA networks within them? Please elaborate more on the consistency and discrepancy of the current studies.

Thanks for your suggestion. We have modified this section and discussed in a more deep way all these aspects, taking in account all the papers in literature and ceRNA networking.

3, In section 7.2, the authors discussed several papers on the prognostic analysis of lncRNAs and OS. Many studies claimed hundreds of lncRNAs have been altered in cancer patients. How would this information be helpful to the treatment of the disease? What would be strategies to use these studies to guide clinical treatment? I look forward to more discussion about this since it is important to transform the knowledge from basic research to clinical application.

Thanks for your important suggestion. We have implemented this part of the review, giving an overview of how daily clinical practice could be ameliorated thanks to the introduction of lncRNAs signatures in the clinical management of pRCC patients

4, Please pay attention to the spelling of some words and make sure they are in modern English. For example, in the abstract, ‘whith’ should be ‘with’, and “hystological” should be “histological”, etc.

Thanks for your suggestion. The manuscript file underwent further revision to track down spelling and eventual grammar errors.

The inclusion of clinical features and treatment strategies in this review is very helpful for understanding the pathology of the disease. With a combination of clinical views and molecular signatures, this review would be beneficial to the scientists and physicians who study pRCC. I suggest a minor revision, and I look forward to seeing the revised manuscript.

Reviewer 2 Report

This review manuscript by Trevisani et al aims to summary the current evidence for the biomarker potentials of long non-coding RNA (lncRNA) in pRCC prognosis.

The review contains detail and update clinical information in the aspects of epidemiology, classification, grading, diagnosis, treatment, and prognosis of pRCC. This section was well written; the updates are useful for the pRCC research community.

However, the clinical section above consists of more than 50% of this review which was supposed to fucus on lncRNA. There are also no clear connections between this clinical section and the following lncRNA section; this review appears to consist of two independent sections. In this regard, its content does not fit the title. This structural deficiency should be corrected.

The lncRNA section also has some structure issues. The authors have briefly represented individual studies without structuring or organizing them with a certain theme. For instance, can these studies be presented according to pathways relevant to pRCC? In the current “collecting setting”, some content was repeatedly presented, for example GAS6-AS1. Some summarizing tables and figures should be prepared for the contents discussed. Key factors in these studies (HR, 90% CI, ROC-AUC, specificity, sensitivity, and p-values) should be included if they are available.

Some statements can be mor precise. Line 236 “… … other RCCs like ccRCC, … …”. As only ccRCC (the main type of RCC) can have worse prognosis than pRCC, I would suggest using ccRCC directly. Line 685 “… …, ROC curves showed a precision higher than 75%… …”; was AUC instead of referred here? Not sure whether AUC can be precisely interpreted as “precision”.

Author Response

Answers to Reviewer 2

This review manuscript by Trevisani et al aims to summary the current evidence for the biomarker potentials of long non-coding RNA (lncRNA) in pRCC prognosis.

The review contains detail and update clinical information in the aspects of epidemiology, classification, grading, diagnosis, treatment, and prognosis of pRCC. This section was well written; the updates are useful for the pRCC research community.

However, the clinical section above consists of more than 50% of this review which was supposed to fucus on lncRNA. There are also no clear connections between this clinical section and the following lncRNA section; this review appears to consist of two independent sections. In this regard, its content does not fit the title. This structural deficiency should be corrected.

Thanks for the valuable hint. Our works usually aim to review current evidence for the clinical potential of biomarker in several type of urinary tract cancers, providing a strong clinical background to fully appreciate and contextualize the molecular medicine discussion, to reach a wider audience. We acknowledge, however, that the initial manuscript file displays the highlighted structural deficiency and, in order to correct it, we tried to synthetize some parts of the clinical section, empowering the following lncRNAs section and harmonizing the different subsections.

The lncRNA section also has some structure issues. The authors have briefly represented individual studies without structuring or organizing them with a certain theme. For instance, can these studies be presented according to pathways relevant to pRCC? In the current “collecting setting”, some content was repeatedly presented, for example GAS6-AS1. Some summarizing tables and figures should be prepared for the contents discussed. Key factors in these studies (HR, 90% CI, ROC-AUC, specificity, sensitivity, and p-values) should be included if they are available.

Thanks for your fundamental suggestion. We have modified the indicated section presenting a more compelling description of lncRNAs biological functions and common pathways. Moreover, we have summarized the most prominent prognostic lncRNAs signatures in a clear and user friendly figure.

Some statements can be mor precise. Line 236 “… … other RCCs like ccRCC, … …”. As only ccRCC (the main type of RCC) can have worse prognosis than pRCC, I would suggest using ccRCC directly. Line 685 “… …, ROC curves showed a precision higher than 75%… …”; was AUC instead of referred here? Not sure whether AUC can be precisely interpreted as “precision”.

Thanks for your suggestion. We corrected the highlighted statements.

Round 2

Reviewer 2 Report

This revised review has satisfactorily addressed my comments and provides comprehensive updates on pRCC particularly in the aspect of long non-coding RNA (lncRNA). This review will be a valuable addition to literature. I support its publication.